# Safety Climate Evaluation in Primary Health Care: A Cross-Sectional Study

**DOI:** 10.3390/ijerph192114344

**Published:** 2022-11-02

**Authors:** Marlene Manuela Moreira Ferreira, Andreia Sofia Costa Teixeira, Tiago Salgado de Magalhães Taveira-Gomes

**Affiliations:** 1Family Health Unit Caldas da Saúde, ACeS Grande Porto I-Santo Tirso/Trofa, 4780-030 Santo Tirso, Portugal; 2Department of Community Medicine, Information and Decision in Health (MEDCIDS), Faculty of Medicine, University of Porto, 4200-450 Porto, Portugal; 3Center for Health Technology and Services Research (CINTESIS), Faculty of Medicine, University of Porto, 4200-450 Porto, Portugal; 4ADiT-LAB, Instituto Politécnico de Viana do Castelo, 4900-367 Viana do Castelo, Portugal; 5Faculty of Health Sciences, University Fernando Pessoa (FCS-UFP), 4200-150 Porto, Portugal; 6MTG Research and Development Lab, 4200-604 Porto, Portugal; 7Toxicology Research Unit (TOXRUN), University Institute of Health Sciences, Advanced Polytechnic and University Cooperative (CESPU), CRL, 4585-116 Gandra, Portugal

**Keywords:** patient safety, primary health care, Safety Attitudes Questionnaire, safety climate, safety culture

## Abstract

The perception of the existence of deficits in patient safety, the associated costs and the limitation of resources have made it essential to define improvement strategies. Important concepts have emerged, such as safety climate, which evaluates the perceptions of safety status held by professionals in relation to their organization. The aim of this study is to characterize the safety climate in primary health care (PHC) using the Safety Attitudes Questionnaire (SAQ)—Short Form 2006 PT and to assess associations between SAQ-Short Form 2006 PT scores and demographic and professional characteristics. A cross-sectional study was conducted in all public PHC units in the northern region of Portugal. Data were collected through an online questionnaire shared via institutional emails and by means of a snowball approach. Descriptive and inferential statistical analysis were performed. Statistical significance set at *p* ≤ 0.05. A total of 649/7427 (8.7%) responses were included in the analyses. The mean and median total SAQ-Short Form 2006 PT scores were 69.23 (SD = 15.73, range 22.22–100.00) and 71.53 [59.03; 79.86], respectively. This is the first study to assess the safety climate in PHC in Portugal. The median obtained total SAQ-Short Form 2006 PT score was 71.53 [59.03; 79.86], which is below the threshold of ≥75, indicating safety deficits.

## 1. Introduction

The first works highlighting the importance of patient safety appeared in the 20th century with the publication of the results of the Harvard Medical Practice Study (1991) [1,2], *To Err is Human: Building a Safer Health System* (2000) [3] and *An organization with a memory* (2002) [4], among others. In 2002, the topic was addressed at the 55th World Health Assembly; in 2004, the World Alliance for Patient Safety was created [5]; and in 2005, the First Global Challenge for Patient Safety was created [6].

The perception of the existence of deficits in patient safety, the associated costs and the limitation of resources have made it essential to define improvement strategies in order to promote safety in association with healthcare delivery [7]. Important concepts have emerged, such as safety culture, which is structural indicator that promotes and facilitates initiatives to minimize risk and prevent adverse events. 

According to the Global patient safety action plan 2021–2030: towards eliminating avoidable harm in health care, “The safety culture of an organization is the product of individual and group values, attitudes, perceptions, competencies, and patterns of behavior that determine the characteristics of the organization’s health and safety management” [8]. 

Safety climate is a more readily measurable component of safety culture. It evaluates the perceptions of safety status held by professionals in relation to their organization [9]. Questionnaires are practical tools that can be used to assess the safety climate, allowing organizations to act proactively and plan improvement strategies, evaluating the impact and effectiveness of implemented actions [7]. 

Research in primary health care (PHC) on the nature, causes or consequences of incidents is performed to a lesser extent than in hospital settings [10]. A meta-analysis published in 2019 revealed pooled prevalence of overall patient harm of 12%, but only two of the studies included in the review where based in PHC setting [11]. This may be due to an assumption that PHC is safer than hospital care and is in the early stages of development in some parts of the world and because records may not be easily accessible, making research in this area difficult [7]. 

A systematic review of PHC safety incidents and their association with harm identified a median of 2–3 incidents per 100 visits/medical records reviewed, 4% of which were associated with severe harm [10]. In PHC, the estimated proportion of patient safety incidents is generally lower than the estimated 10% of people who experience events in hospitals. However it is important to consider that in many parts of the world, the overall volume of people using PHC is substantially higher than those using hospital services. Therefore, the burden of potential harm related to patient safety incidents in PHC cannot be underestimated [10].

A recent study analyzed 1129 patient safety incidents reports in PHC and found that 25.0% did not reach the patient, 66.5% reached the patient without harm and 8.5% caused adverse events. By analyzing adverse events, it was possible to verify that 46.9% required observation, 34.4% caused temporary damage and required treatment, 13.5% required hospitalization and 5.2% caused severe permanent damage and/or a situation close to death. In this study, the main identified critical areas were communication (27.8%), management (25.1%), care delivery (23.5%) and medication (18.4%); few incidents were related to diagnosis (3.6%). Concerning specifically adverse events, the main identified critical areas were care delivery (33.3%), communication (20.8%), management (17.7%) and medication (15.6%); diagnosis was the least represented area (11.5%). The main contributing factors were related to the professional (55.8%), the organization (36.8%) and external factors (25.9%). For adverse events, the contributing factors were also related to the professional (47.9%), the organization (40.6%) and external factors (33.3%) [12].

Safety and quality in health care are linked; identification of critical points that require intervention, once flagged, allows for the development and implementation of improvement strategies [13]. 

The aim of this study was to characterize the safety climate in public PHC in the northern region of Portugal using the SAQ-Short Form 2006 PT and to assess associations between SAQ-Short Form 2006 PT scores (total and by domain) and demographic and professional characteristics.

## 2. Materials and Methods

### 2.1. Study Design

A cross-sectional study was conducted in all public PHC units in the northern region of Portugal. 

### 2.2. Setting and Subjects

In mainland Portugal, public PHC is structured according to five regional health administrations (Alentejo, Algarve, Center, Lisboa e Vale do Tejo and North), subdivided into health center clusters (ACeS), including, among others, PHC units with varying types of organization and remuneration (family health units A (USF-A), family health units B (USF-B) and customized healthcare units (UCSP)) [14]. 

The research population in the present study included all physicians, doctors in pre-career training, nurses and technical assistants working in PHC units in the North Regional Health Administration in Portugal in 2022. According to Bi-CSP^®^ data (bicsp.min-saude.pt, accessed on 31 January 2022), this population includes a total of 7427 professionals (Appendix A): 2318 physicians (31.2%), 929 doctors in pre-career training (12.5%), 2386 nurses (32.1%) and 1794 technical assistants (24.2%). 

Professionals who refused to participate or who did not met the eligibility criteria (first-time participants in the study who worked as physicians, doctors in pre-career training, nurses and technical assistants in PHC units in the North Regional Health Administration in Portugal at their current workplace for a month or more) were excluded.

### 2.3. Data Collection

Data were collected from 7 June to 9 August 2022 through an online structured Google Forms^®^ questionnaire shared between 7 June and 5 August 2022 via institutional emails and by means of a snowball approach, i.e., a non-probabilistic sampling whereby participants help spread the word, recruiting additional respondents. The questionnaire was voluntary, self-completed and totally anonymous. It included an introductory message, informed consent to use data for the research purposes, questions about basic demographic and professional characteristics and the Safety Attitudes Questionnaire—Short Form 2006 validated for Portugal (SAQ—Short Form 2006 PT, Appendix B) [7]. A pilot test was conducted prior to the release of the questionnaire to participants. Completion of all items was defined as mandatory for response submission.

### 2.4. Variables and Instruments

The SAQ, developed by Sexton et al., is a widely used tools that has been adapted for use in intensive care units, operating rooms, and general inpatient and ambulatory clinics [7,9] (Appendix C). It has been extensively tested, is considered psychometrically valid and allows for the assessment of the perceptions and attitudes of health professionals related to safety [7]. The original version was developed in 2006 by Bryan Sexton, Eric Thomas and Bob Helmreich (University of Texas) based on the Intensive Care Unit Management Attitudes Questionnaire, which was derived from the Flight Management Attitudes Questionnaire, which is widely used in commercial aviation [9].

The SAQ—Short Form 2006 was translated, culturally adapted and validated for Portugal by Saraiva et al. [7] from the original SAQ version developed by Sexton et al. [9]. Permission to use this instrument in the present study was obtained from the copyright holder of the instrument by the one of the authors. This questionnaire assesses organizational, work environment-related and team-related factors [7]. 

The SAQ-Short Form 2006 PT consists of two parts [7]: 

The first part of the SAQ-Short Form 2006 PT consists of 41 items (Total SAQ Cronbach’s alpha (α): 0.92) and six domains: *team environment* (perception of the quality of collaboration among professionals; items 1–6; α = 0.70), *safety climate* (perception of a solid and proactive organizational commitment to safety; items 7–13; α = 0.73), *job satisfaction* (pleasant or emotionally positive feeling resulting from the perception of the work experience; items 15–19; α = 0.86), *stress recognition* (recognition of how performance is influenced by stressful factors; items 20–23; α = 0.82), *management perception* (management approval regarding safety issues; unit and hospital; items 24–29; α = 0.88) and *working conditions* (perception of quality environment and logistics related to the workplace; items 30–32; α = 0.71). Items 14 and 33–36 are not part of the aforementioned domains, and items 2, 11 and 36 are scored inversely. In order to improve completion of the questionnaire by PHC professionals, three references were added to the SAQ-Short Form 2006 PT (Google Forms version), defining equivalence between terms: hospital = ACeS; unit = UCSP/USF; and clinical area = UCSP/USF. The answers to each item were provided on a Likert scale consisting of five levels: 1 = strongly disagree; 2 = partially disagree; 3 = neutral; 4 = partially agree; 5 = strongly agree; X = not applicable [7,9]. A calculation formula suggested by the authors of the original study [9], was applied, which involved averaging the responses (after inversion of reverse items 2, 11 and 36) and to calculation of a 100-point scale score (1 was subtracted from the mean, and the result was multiplied by 25). The final score ranges from 0–100 (0 = worst perception of the safety climate; 100 = best perception of the safety climate). “Not applicable” responses were considered missing [7,9]. A score of ≥75 indicates a strong safety environment, reflecting positive attitudes toward patient safety [13]. 

The second part of the questionnaire includes information on demographic and professional characteristics (*gender*, *function* and *length of service*) [7]. *Age*, *workplace* and *type of primary care unit* were added.

### 2.5. Data Analysis

Categorical variables were described by absolute and relative frequencies (*n* (%)). Normally distributed continuous variables were summarized using means and standard deviation (M (SD)). Non-normally distributed continuous variables were described using median and interquartile interval (Med [Q_1_; Q_3_]), where Q_1_ is the first quartile, and Q_3_ the third quartile. In ordinal variables, although it is recommended to present Med [Q_1_; Q_3_], we decided to also present M (SD) to facilitate interpretation and possible comparisons with the results of other studies. The normality of distributions was verified by visual observation of histograms and using the Kolmogorov–Smirnov test. 

The internal consistency of the SAQ-Short Form 2006 PT scale and its six domains was assessed using Cronbach’s alpha (α), with values above 0.70 considered acceptable. To study the association between two continuous variables, Pearson’s coefficient (rP) was used when the two variables were normally distributed or Spearman’s coefficient (rS) when at least one of the variables was not normally distributed or was ordinal. To compare means between 2 independent, normally distributed samples, a t-test for independent samples was used. To compare distributions between 2 independent, not normally distributed samples, the Mann–Whitney test was used. To compare means between 3 or more independent, normally distributed samples, one-way ANOVA test was used. To compare distributions between 3 or more independent, not normally distributed samples, the Kruskal–Wallis test was used. In both the one-way ANOVA test and the Kruskal–Wallis test, when significant differences were found, multiple comparisons were performed with Bonferroni adjustments. 

The Statistical Package for the Social Sciences^®^ (SPSS, version 27.0 for Windows^®^ SPSS Inc., Chicago, IL, USA, 1989–2020) to analyze the results, and statistical significance was set at *p* ≤ 0.05.

### 2.6. Ethical Considerations

This study was authorized by the management boards of the institutions involved and approved by four ethics committees of the northern region of Portugal related to PHC (ethics committees of the North Regional Health Administration, the Alto Minho Local Health Unit, the Matosinhos Local Health Unit and the Northeast Local Health Unit). To avoid loss of anonymity, the names of PHC units were not collected. All participants signed an informed consent form. All recommended ethical principles were respected.

## 3. Results

### 3.1. Descriptive Analysis of the Sample

A total of 676/7427 (9.1%) responses were collected. Among these, 2 (2/676, 0.3%) were excluded because the respondents did not consent to participate, and 25 (25/676, 3.7%) were excluded because the respondents did not meet the eligibility criteria (first time participants in the study who worked as physicians, doctors in pre-career training, nurses and technical assistants in PHC units in the North Regional Health Administration in Portugal at their current workplace for a month or more). The sample was stratified by *function* and *workplace*.

A total of 649/7427 (8.7%) responses were included in the analyses. The mean age of respondents was 42.6 years (SD = 9.9, range 18–67 years), and 77% (500/649) were women. The following distribution was found in terms of demographic and professional characteristics: (a) *age*: 0.2% (1/649) 18–20 years; 9.7% (63/649) 21–30 years; 37.1% (241/649) 31–40 years; 34.1% (221/649) 41–50 years; 11.6% (75/649) 51–60 years; 7.4% (48/649) 61–70 years; (b) *function*: 21.9% (142/649) were technical assistants; 31.7% (206/649) were nurses; 10.9% (71/649) were doctors in pre-career training; and 35.4% (230/649) were physicians; (c) *length of service at the current workplace*: 2.5% (16/649), 1–5 months; 8.5% (55/649), 6–11 months; 13.7% (89/649), 1–2 years; 12.2% (79/649), 3–4 years; 13.7% (89/649), 5–10 years; 29.0% (188/649), 11–20 years; 20.5% (133/649), 21 or more years; (d) *workplace* (Table 1); (e) *type of primary care unit*: 19.7% (128/649), UCSP; 22.5% (146/649), USF-A; 57.8% (375/649), USF-B.

### 3.2. Descriptive Analysis of SAQ—Short Form 2006 PT

A descriptive analysis of the SAQ-Short Form 2006 PT was performed. All 41 items of the SAQ-Short Form 2006 PT were analyzed. The median response rate for “not applicable” considering the totality of the items of the SAQ-Short Form 2006 PT was 3.2%. These responses were excluded from the analysis.

In order to evaluate the internal consistency of the present survey, Cronbach’s alpha (α) was determined: total SAQ-Short Form 2006 PT, α = 0.93; *team environment* (items 1–6), α = 0.79; *safety climate* (items 7–13), α = 0.82; *job satisfaction* (items 15–19), α = 0.91; *stress recognition* (items 20–23), α = 0.88; *management perception* (items 24–29), α = 0.89; and *working conditions* (items 30–32), α = 0.72.

The mean and median total SAQ-Short Form 2006 PT scores were 69.23 (SD = 15.73, range 22.22–100.00) and 71.53 [59.03; 79.86], respectively (Table 2). 

The median SAQ-Short Form 2006 PT scores by domain ranged from 54.55 [38.64; 70.45] to 93.75 [75.00; 100.00] (Table 2). *Stress recognition* obtained the highest median score of 93.75 [75.00; 100.00], and *management perception* and *working conditions* were associated with the lowest median scores of 54.55 [38.64; 70.45] and 66.67 [50.00; 83.33], respectively (Table 2).

The median scores obtained in the *management perception* domain differed when calculated for UCSP/USF and ACeS: 62.50 [41.67; 75.00] and 50.00 [30.21; 66.67], respectively.

Item 7 had the highest mean score of 4.47 (SD = 0.92) (median score: 5 [4; 5]), and item 24b obtained the lowest mean score of 2.66 (SD = 1.33) (median score: 3 [1; 4]) (Table 3). 

Four items obtained mean scores of less than 3 (Table 3): item 36: mean score of 2.93 (SD = 1.31) (median score: 3 [2; 4]); item 29: mean score of 2.90 (SD = 1.48) (median score of 3 [2; 4]); item 27b: mean score of 2.79 (SD = 1.25) (median score of 3 [2; 4]); and item 24b: mean score of 2.66 (SD = 1.33) (median score of 3 [1; 4]).

Figure 1 shows the distribution of responses for each of the 41 items of the SAQ—Short Form 2006 PT according to the Likert scale. These are constructed using absolute and relative frequencies (*n* (%)) without inversion of items 2, 11 and 36 and after exclusion of “not applicable” responses. The total percentage of “not applicable” responses was 7.0%. 

Item 35 received the highest percentage of “indifferent” responses. Item 34 received the highest percentage of positive responses, i.e., “Agree” or “Strongly agree”. Items 2 and 11 obtained the highest percentage of negative responses, i.e., “Disagree” or “Strongly disagree”. Item 24b obtained the third-highest percentage of negative responses, i.e., “Disagree” or “Strongly disagree”. Some of the items included in the *management perception* domain (items 24–29) were among those with highest percentage of negative answers, i.e., “Disagree” or “Strongly disagree”.

### 3.3. Inferential Analysis

Each of the domains and the total SAQ-Short Form 2006 PT scores (Table 4) are described below.


*Team environment*
The team environment domain is only associated with the characteristic function (*p* < 0.001) and type of primary care unit (*p* < 0.001). Regarding the variable function, multiple comparisons with Bonferroni adjustments show that differences were found between the categories of nurses and physicians (*p* < 0.001) and between nurses and doctors in pre-career training (*p* = 0.004), with the nurse category showing lower values in this domain, as well as between the categories of technical assistants and physicians (*p* = 0.007) and between technical assistants and doctors in pre-career training (*p* = 0.015), with the technical assistant category also showing lower values for this domain. With respect to the variable type of primary care unit, multiple comparisons with Bonferroni adjustments revealed differences between the categories of UCSP and USF-A (*p* < 0.001) and between UCSP and USF-B (*p* < 0.001), with the UCSP category showing lower values for this domain.Safety climateThe safety climate domain is only associated with the type of primary care unit variable (*p* < 0.001). Multiple comparisons with Bonferroni adjustments revealed differences between the categories of UCSP and USF-A (*p* < 0.001) and between UCSP and USF-B (*p* < 0.001), with the UCSP category showing lower values for this domain.Job satisfactionThe job satisfaction domain is associated with gender (*p* = 0.001) and type of primary care unit (*p* < 0.001). With regard to gender, women show higher values in this domain than men. With respect to the type of primary care unit variable, multiple comparisons with Bonferroni adjustments revealed differences between the categories of UCSP and USF-B (*p* < 0.001), with the UCSP category showing lower values for this domain.Stress recognitionThe stress recognition domain shows associations with the characteristics of age (*p* < 0.001), length of service at the current workplace (*p* = 0.032) and function (*p* < 0.001). As the age of the respondent increases, the score obtained in this domain decreases. However, the linear correlation is weak (rS = −0.166). As the length of service at the respondent’s current workplace increases, so does the score obtained in this domain. However, this linear correlation is also weak (rS = −0.095). With respect to the variable function, multiple comparisons with Bonferroni adjustments revealed differences between the categories of technical assistants and nurses (*p* < 0.001), between technical assistant and doctors in pre-career training (*p* < 0.001) and between technical assistants and physicians (*p* < 0.001), with the technical assistant category showing lower values for this domain, as well as between the categories of nurses and physicians (*p* = 0.005), with the nurse category showing lower values for this domain than physicians. Management perceptionThe management perception domain is associated with gender (*p* = 0.024) and type of primary care unit (*p* = 0.038). With respect to gender, women show higher values in this domain than men. Regarding the type of primary care unit variable, multiple comparisons with Bonferroni adjustments revealed differences between the categories of UCSP and USF-B (*p* = 0.044), with the UCSP category showing lower values for this domain.Working conditionsThe working conditions domain is associated with the characteristics function (*p* = 0.016) and type of primary care unit (*p* < 0.001). With respect to the function variable, multiple comparisons with Bonferroni adjustments revealed differences between the categories of technical assistants and physicians (*p* = −0.014) and between technical assistants and doctors in pre-career training (*p* = 0.048), with the technical assistant category showing lower values for this domain. With respect to the type of primary care unit variable, multiple comparisons with Bonferroni adjustments revealed differences between the categories of UCSP and USF-A (*p* < 0.001) and between UCSP and USF-B (*p* < 0.001), with the UCSP category presenting lower values for this domain.Total SAQ-Short Form 2006 PT scoreThe total SAQ-Short Form 2006 PT score shows association with the gender (*p* = 0.019) and type of primary care unit (*p* < 0.001). With respect to gender, women show higher values than men in the total score. With respect to the type of primary care unit variable, multiple comparisons with Bonferroni adjustments revealed differences between the categories UCSP and USF-A (*p* < 0.001) and between UCSP and USF-B (*p* < 0.001), with the UCSP category presenting lower total scores. 

As shown in Figure 2, the ACeS is associated with the domains *team environment* (*p* < 0.001), *safety climate* (*p* < 0.001), *job satisfaction* (*p* = 0.042), *management perception* (*p* = 0.011), *working conditions* (*p* < 0.001) and with the Total SAQ-Short Form 2006 PT score (*p* = 0.009). 

Multiple comparisons with Bonferroni adjustments were performed. The differences found for each of the domains considering the *workplace* variable are shown in Figure 2 and presented in the following sections. 


*Team environment*
“4. ACeS Alto Trás-os-Montes—Nordeste” and “17. ACeS Grande Porto V—Porto Ocidental” (*p* = 0.007); “4. ACeS Alto Trás-os-Montes—Nordeste” and “21. ACeS Matosinhos” (*p* = 0.010); “4. ACeS Alto Trás-os-Montes—Nordeste” and “8. ACeS Cávado III—Barcelos/Esposende” (*p* = 0.010); “4. ACeS Alto Trás-os-Montes—Nordeste” and “7. ACeS Cávado II—Gerês/Cabreira” (*p* < 0.001); “22. ACeS Tâmega I—Baixo Tâmega” and “7. ACeS Cávado II—Gerês/Cabreira” (*p* = 0.002) and “23. ACeS Tâmega II—Vale do Sousa Sul” and “7. ACeS Cávado II—Gerês/Cabreira” (*p* = 0.019).Safety climate“4. ACeS Alto Trás-os-Montes—Nordeste” and “13. ACeS Grande Porto I—Santo Tirso/Trofa” (*p* = 0.007) and “4. ACeS Alto Trás-os-Montes—Nordeste” and “7. ACeS Cávado II—Gerês/Cabreira” (*p* = 0.001).Job satisfaction“4. ACeS Alto Trás-os-Montes—Nordeste” and “13. ACeS Grande Porto I—Santo Tirso/Trofa” (*p* = 0.023).Management perceptionNo differences were found between pairs of categories.Working conditions“22. ACeS Tâmega I—Baixo Tâmega” and “16. ACeS Grande Porto IV—Póvoa do Varzim/Vila do Conde” (*p* = 0.027) and “22. ACeS Tâmega I—Baixo Tâmega” and “17. ACeS Grande Porto V—Porto Ocidental” (*p* = 0.010).Total SAQ-Short Form 2006 PT score“4. ACeS Alto Trás-os-Montes—Nordeste” and “13. ACeS Grande Porto I—Santo Tirso/Trofa” (*p* = 0.040).

## 4. Discussion

### 4.1. Principal Findings

This was the first study to evaluate the safety climate in PHC in Portugal. A score ≥75 indicates a strong safety environment [13], a cutoff value not accomplished in the present study, as the mean and median SAQ-Short Form 2006 PT total scores were 69.23 (SD = 15.73, range: 22.22–100.00) and 71.53 [59.03; 79.86], respectively.

The total SAQ-Short Form 2006 PT median scores were higher among female respondents, in *workplace* “13. ACeS Grande Porto I—Santo Tirso/Trofa” and in *type of primary care unit* USF-A and USF-B. In contrast, the lowest median values among male respondents, in “4. ACeS Alto Trás-os-Montes—Nordeste” and in UCSP. The *workplace* “13. ACeS Grande Porto I—Santo Tirso/Trofa” was considered, in recent, one of the best in terms of performance in North Regional Health Administration (Bi-CSP^®^: bicsp.min-saude.pt, access in 14 September 2022), which might be related to safety perception. “4. ACeS Alto Trás-os-Montes—Nordeste” obtained a lower score in terms of performance in the North Regional Health Administration, although it was not the *workplace* with the lowest rate (Bi-CSP^®^: bicsp.min-saude.pt, accessed on 14 September 2022). These results suggest an association between the safety climate and the level of performance, although such an association remains to be confirmed. USF-A and USF-B have organizational, functional and technical autonomy, which might explain the achievement of higher total SAQ-Short Form 2006 PT scores in comparison to the UCSP.

The scores obtained in the domains of *team environment*, *safety climate*, *job satisfaction* and *stress recognition* indicate a strong safety environment. In contrast, the scores obtained in the domains of *management perception* and *working conditions* (below the threshold of 75) indicate the need for improvement. 

The differences obtained in SAQ-Short Form 2006 PT domain scores identify areas that require further intervention, which are discussed in the following sections.


*Team environment*
The lowest scores in the team environment domain were obtained for the categories of nurse, technical assistant and UCSP, indicating the need for investment in activities that promote communication skills and teamwork, especially with nurses, technical assistants and in the UCSP.Safety climateThe lowest median score in the safety climate domain was obtained in the UCSP, possible because that USF-A and USF-B have established, as a goal, the sharing and discussion of incidents at meetings. Information should be shared in an open, non-punitive environment and on a regular basis.Job satisfactionThe lowest scores in the job satisfaction domain were obtained among male respondents and in the UCSP. The USF-A and USF-B have, as previously mentioned, organizational, functional and technical autonomy, which might increase job satisfaction and contribute to professional fulfillment. Furthermore, professionals working in the USF-B have higher salaries compared to those employed by the UCSP and USF-A, likely explaining the obtained results.Stress recognitionIn the stress recognition domain, as the age of the respondent increases, the obtained SAQ-Short Form 2006 PT median score decreases, and as the length of service at the respondent’s current workplace increases, so does the obtained score. The lowest scores in *n* this domain were associated with the categories of nurse and technical assistant. Healthcare professionals should be able to recognize excessive workload and fatigue in order to improve the safety climate and promote effectiveness. Keeping this concept in mind is especially important as the age of professionals increases, with shorter length of service at the current workplace and in the categories of nurse and technical assistant. Self-care might also be relevant to this domain and should be promoted.Management perceptionThe score obtained in the management perception domain was below the threshold of 75, indicating the need for improvement. The lowest median scores in this domain were obtained among male respndents and in the UCSP. The UCSP has less autonomy than the USF-A and USF-B, which might influence management perception. The implementation of measures that promote the transition from the UCSP to the USF might improve this result.The median score in the management perception domain was also calculated separately for ACeS and UCSP/USF, with lower values associated with ACeS management. In the USF, management is not hierarchical and develops in a cooperative atmosphere witch might have contributed to the better result obtained in the UCSP/USF. Healthcare professionals perceive that managers of ACeS and UCSP/USF need to improve in terms of compromise of patient safety, management performance, support of professional efforts, constructive personnel problem solving and ensuring appropriate human resources. The lack of appropriate human resources is particularly evident, with a negative score for this item (29). The need for improvement with respect to support of professional efforts and constructive personnel problem solving is particulary evident for ACeS management, with negative mean scores for these items (24b,e and 27b). This domain is in need of attention from managers and intervention.Working conditionsThe SAQ-Short Form 2006 PT median score obtained in the working conditions domain was also below the threshold of 75, indicating the need for improvement. In this domain, low scores in the categories of technical assistant and UCSP indicate the need for improvement in personnel training and appropriate clinical decision support tools. 

The category of UCSP obtained low median scores in several domains: *team environment*, *safety climate*, *job satisfaction*, *management perception* and *working conditions*. According to Bi-CSP^®^ data (bicsp.min-saude.pt, accessed on 31 January 2022), the *workplace* “4. ACeS Alto Trás-os-Montes—Nordeste” is the only one in the North Regional Health Administration totally constituted by UCSP (Appendix D) and was identified in the present study as having the lowest total SAQ-Short Form 2006 PT score. This does not reflect all variables inherent to each *workplace*, nor the quality of work developed.

Item 7 obtained the highest mean score, reflecting a positive perception of professionals with respect to patient care; item 24b obtained the lowest mean score, reflecting a perceived need for increased support from ACeS leadership.

Four items were obtained mean scores of less than 3: “36” is an inverse, but still with low scores, indicating a need for improved communication; “29”, reflecting the perception of an insufficient ratio of professionals with respect to the number of patients; and “27b” and “24b”, reflecting a perceived need for improvement in management of problems related to professionals and more support from ACeS leadership.

Item 35 received the highest percentage of “indifferent” responses. These results might be explained by difficulties in interpreting the question, as in most situations, pharmacists are not integrated in PHCs but work in community pharmacies. Item 34 received highest percentage of positive responses, i.e., “Agree” or “Strongly agree”, indicating a positive perception of collaboration with staff physicians. Items 2 and 11 receieved the highest percentage of negative responses, i.e., “Disagree” or “Strongly disagree”, a result to be inversed but still with potential for improvement. Item 24b received the third-highest percentage of negative responses, i.e., “Disagree” or “Strongly disagree”, highlighting the perception of a need for more support from ACeS management, as previously mentioned.

Some of the items included in the *management perception* domain (items 24–29) are among those with a high percentage of negative answers, i.e., “Disagree” or “Strongly disagree”, calling for leadership investment in communication, problem solving and professional support.

### 4.2. Comparison with Prior Work

Several papers have been published to date assessing safety climate. However, comparative analysis is limited because most such studies were conducted in hospitals or with different study populations, with some using a different assessment instrument other than the SAQ-Short Form 2006. Another limitation has to do with the differences between the validated SAQ-Short Form 2006 questionnaires used in different countries, as the items included in each domain are not always the same.

In 2006, Sexton et al. conducted a study in ambulatory clinics in the USA and obtained lower mean scores than those reported in the present study: *team environment*, 69.7 (vs. 77.41% in the present study); *safety climate*, 69.9 (vs. 71.91 in the present study); *job satisfaction*, 70.6 (vs. 75.61 in the present study); *stress recognition*, 66.7 (vs. 83.42 in the present study); and *working conditions*, 61.6 (vs. 66.55 in the present study); with the exception of *perceptions of management*, 55.3 (vs. 54.66 in the present study) [9].

In 2010, McGuire et al. conducted a study in a large practice with a primary care core in the USA staffed by clinicians in family medicine, pediatrics and internal medicine and obtained higher mean scores than those reported in the present study: *team environment*, 88.9 (vs. 77.41 in the present study); *safety climate*, 87.8 (vs. 71.91 in the present study); *job satisfaction*, 86.2 (vs. 75.61 in the present study); *stress recognition*, 74.8 (vs. 83.42 in the present study); perceptions of *executive management*, 72.6; perceptions of *local management*, 86.0 (vs. *management perception* 54.66 in the present study); and *working conditions*, 84.9 (vs. 66.55 in the present study) [15]. 

In 2019, Lousada et al. conducted a study in six primary care centers in Brazil and obtained lower mean scores than those reported in the present study: total SAQ-Short Form 2006 score, 62.6 (vs. 69.23 in the present study); *team environment*, 69.4 (vs. 77.41 in the present study); *safety climate*, 59.5 (vs. 71.91 in the present study); *job satisfaction*, 75.1 (vs. 75.61 in the present study); *stress recognition*, 65.8 (vs. 83.42 in the present study); *management perception*, 54.5 (vs. 54.66 in the present study); and *working conditions*, 51.2 (vs. 66.55 in the present study) [16]. 

Differences between studies are expected, as safety climate evaluates the perceptions of safety status held by professionals in relation to their organization [9]. 

### 4.3. Strengths and Limitations

This is the first study to evaluate the safety climate in PHC in Portugal. We intended for the study to be comprehensive, involving all public PHC units of the North Regional Health Administration, and inclusive, with the participation of physicians, doctors in pre-career training, nurses and technical assistants. The obtained sample was representative of the studied population. The SAQ questionnaire is a widely used instrument with good psychometric qualities, and the consistency of the data gathered in this study supports its validity. By identifying multiple domains in safety climate, the SAQ highlights critical points and possible areas of intervention. 

With respect to limitations, this study is a self-reported survey depending on the respondents’ recall, meaning that results might be affected by reporting bias. The opportunity to participate in the study by filling in a form exclusively available online may represent a barrier to participation. The results may reflect, at least partially, effects of the COVID-19 pandemic. The cross-sectional nature of the study means the results do not measure any evolution that may have occurred. In this study, we also analyzed only a part of safety culture, focusing on professionals’ perceptions.

Studies in this area are still scarce, indicating the need for attention from researchers.

## 5. Conclusions

This is the first study to assess the safety climate in PHC in Portugal. The median total SAQ-Short Form 2006 PT score was 71.53 [59.03; 79.86], which is below the threshold of ≥75, indicating safety deficits. The lowest median scores were obtained in the domains of *management perception* and *working conditions*. Further studies are needed to determine which factors might influence the safety climate in a positive or in a negative manner, as well as associations with performance and the occurrence of incidents. 

## Figures and Tables

**Figure 1 ijerph-19-14344-f001:**
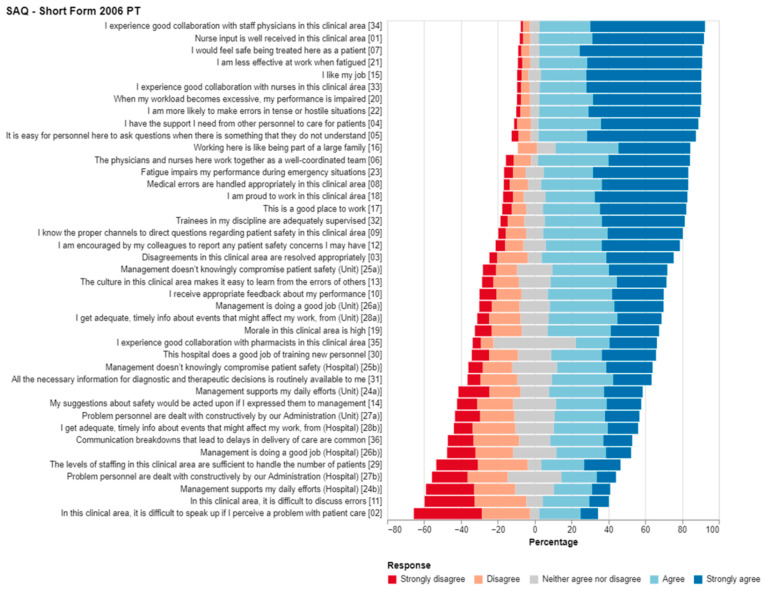
Descriptive Analysis of the SAQ—Short Form 2006 PT per item. The 41 SAQ-Short Form 2006 PT items and respective Likert scale responses are presented (without inversion of items 2, 11 and 36). In order to improve the completion of the questionnaire by PHC professionals, three references were added to SAQ-Short Form 2006 PT (Google Forms version), defining equivalence between terms: hospital = ACeS, unit = UCSP/USF and clinical area = UCSP/USF.

**Figure 2 ijerph-19-14344-f002:**
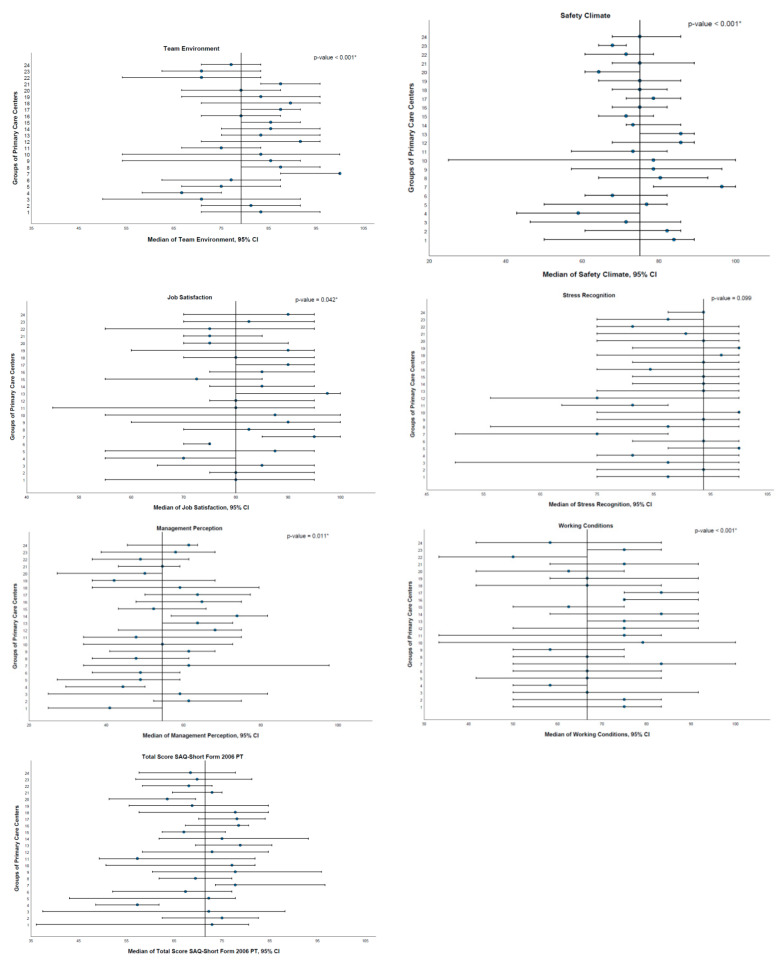
Medians and respective 95% CI of the six domains and total SAQ-Short Form 2006 PT score by ACeS. The Kruskal–Wallis test was used. *: significant at 5%.

**Table 1 ijerph-19-14344-t001:** Descriptive analysis of the sample: *workplace* and *function*.

Workplace	Function					
	Physicians *n*	Doctors in Pre-Career Training *n*	Nurse *n*	Technical Assistant *n*	Total	
					*n*	%
1. ACeS Alto Ave—Guimarães, Vizela e Terras de Basto	12	4	10	6	32	4.9
2. ACeS Alto Minho	12	3	12	8	35	5.4
3. ACeS Alto Trás-os-Montes—Alto Tâmega e Barroso	5	3	6	5	19	2.9
4. ACES Alto Trás-os-Montes—Nordeste	9	3	9	16	37	5.7
5. ACeS Ave/Famalicão	9	4	4	4	21	3.2
6. ACeS Cávado I—Braga	15	6	12	7	40	6.2
7. ACeS Cávado II—Gerês/Cabreira	4	1	7	4	16	2.5
8. ACeS Cávado III—Barcelos/Esposende	7	4	10	6	27	4.2
9. ACeS Douro I—Marão e Douro Norte	7	1	6	3	17	2.6
10. ACeS Douro II—Douro Sul	2	1	4	3	10	1.5
11. ACeS Entre Douro e Vouga I—Feira e Arouca	5	3	7	8	23	3.5
12. ACeS Entre Douro e Vouga II—Aveiro Norte	5	3	4	7	19	2.9
13. ACeS Grande Porto I—Santo Tirso/Trofa	14	2	8	4	28	4.3
14. ACeS Grande Porto II—Gondomar	8	4	5	4	21	3.2
15. ACeS Grande Porto III—Maia/Valongo	24	4	7	11	46	7.1
16. ACeS Grande Porto IV—Póvoa do Varzim/Vila do Conde	7	4	15	4	30	4.6
17. ACeS Grande Porto V—Porto Ocidental	20	4	9	4	37	5.7
18. ACeS Grande Porto VI—Porto Oriental	7	3	5	4	19	2.9
19. ACeS Grande Porto VII—Gaia	10	4	6	3	23	3.5
20. ACeS Grande Porto VIII—Espinho/Gaia	10	2	8	5	25	3.9
21. ACeS Matosinhos	15	2	8	4	29	4.5
22. ACeS Tâmega I—Baixo Tâmega	8	1	15	9	33	5.1
23. ACeS Tâmega II—Vale do Sousa Sul	6	4	8	8	26	4.0
24. ACeS Tâmega III—Vale do Sousa Norte	9	1	21	5	36	5.5
TOTAL	230	71	206	142	649	100.0

**Table 2 ijerph-19-14344-t002:** Descriptive analysis of the SAQ—Short Form 2006 PT: mean (SD); median [Q1; Q3] of total score and per domain.

SAQ—Short Form 2006 PT	*n*	Mean (SD)	Median [Q_1_; Q_3_]
Total score	350	69.23 (15.73)	71.53 [59.03; 79.86]
*Team environment* (items 1–6)	608	77.41 (19.38)	79.17 [66.67; 91.67]
*Safety climate* (items 7–13)	587	71.91 (20.00)	75.00 [60.71; 85.71]
*Job satisfaction* (items 15–19)	597	75.61 (24.13)	80.00 [60.00; 95.00]
*Stress recognition* (items 20–23)	503	83.42 (21.34)	93.75 [75.00; 100.00]
*Management perception* (items 24–29)	486	54.66 (22.20)	54.55 [38.64; 70.45]
*Working conditions* (items 30–32)	492	66.55 (23.97)	66.67 [50.00; 83.33]

**Table 3 ijerph-19-14344-t003:** Descriptive analysis of the SAQ—Short Form 2006 PT: mean (SD); median [Q_1_; Q_3_] per item.

SAQ-Short Form 2006 PT, per Item	Mean (SD)	Median [Q_1_; Q_3_]
***Team environment* (items 1–6)**		
1. Nurse input is well received in this clinical area. (^a^ clinical area = UCSP/USF), *n* = 636	4.42 (0.89)	5 [4; 5]
2. In this clinical area, it is difficult to speak up if I perceive a problem with patient careb. (^a^ clinical area = UCSP/USF), *n* = 640	3.59 (1.41)	4 [2; 5]
3. Disagreements in this clinical area are resolved appropriately (i.e., not who is right, but what is best for the patient). (^a^ clinical area = UCSP/USF), *n* = 638	3.83 (1.21)	4 [3; 5]
4. I have the support I need from other personnel to care for patients, *n* = 636	4.29 (0.97)	5 [4; 5]
5. It is easy for personnel here to ask questions when there is something that they do not understand. (^a^ Unit = UCSP/USF), *n* = 634	4.30 (1.07)	5 [4; 5]
6. The physicians and nurses here work together as a well-coordinated team. (^a^ Unit = UCSP/USF), *n* = 639	4.09 (1.11)	4 [4; 5]
***Safety climate* (items 7–13)**		
7. I would feel safe being treated here as a patient, *n* = 629	4.47 (0.92)	5 [4; 5]
8. Medical errors are handled appropriately in this clinical area. (^a^ clinical area = UCSP/USF), *n* = 631	4.10 (1.10)	4 [4; 5]
9. I know the proper channels to direct questions regarding patient safety in this clinical area. (^a^ clinical area = UCSP/USF), *n* = 632	3.97 (1.14)	4 [4; 5]
10. I receive appropriate feedback about my performance, *n* = 642	3.59 (1.28)	4 [3; 5]
11. In this clinical area, it is difficult to discuss errors b. (^a^ clinical area = UCSP/USF), *n* = 640	3.36 (1.38)	4 [2; 5]
12. I am encouraged by my colleagues to report any patient safety concerns I may have, *n* = 638	3.95 (1.19)	4 [3; 5]
13. The culture in this clinical area makes it easy to learn from the errors of others. (^a^ clinical area = UCSP/USF), *n* = 632	3.64 (1.19)	4 [3; 5]
***Single item* (item 14)**		
14. My suggestions about safety would be acted upon if I expressed them to management, *n* = 626	3.23 (1.27)	3 [2; 4]
***Job satisfaction* (items 15–19)**		
15. I like my job, *n* = 607	4.41 (0.95)	5 [4; 5]
16. Working here is like being part of a large family, *n* = 632	3.89 (1.21)	4 [3; 5]
17. This is a good place to work, *n* = 632	4.06 (1.16)	4 [4; 5]
18. I am proud to work in this clinical area. (^a^ clinical area = UCSP/USF), *n* = 626	4.11 (1.15)	5 [4; 5]
19. Morale in this clinical area is high, *n* = 638	3.52 (1.28)	4 [2; 5]
***Stress recognition* (items 20–23)**		
20. When my workload becomes excessive, my performance is impaired, *n* = 603	4.37 (0.95)	5 [4; 5]
21. I am less effective at work when fatigued, *n* = 602	4.42 (0.94)	5 [4; 5]
22. I am more likely to make errors in tense or hostile situations, *n* = 602	4.38 (0.97)	5 [4; 5]
23. Fatigue impairs my performance during emergency situations (e.g., emergency resuscitation, seizure), *n* = 530	4.14 (1.15)	5 [4; 5]
***Management perception* (items 24–29)**		
24. (a) Management supports my daily efforts (Unit Management) (^a^ Unit = UCSP/USF), *n* = 635	3.21 (1.39)	4 [2; 4]
24. (b) Management supports my daily efforts (Hospital Administration). (^a^ hospital = ACeS), *n* = 561	2.66 (1.33)	3 [1; 4]
25. (a) Management doesn’t knowingly compromise patient safety (Unit Management) (^a^ Unit = UCSP/USF), *n* = 621	3.68 (1.23)	4 [3; 5]
25. (b) Management doesn’t knowingly compromise patient safety (Hospital Administration). (^a^ hospital = ACeS), *n* = 554	3.45 (1.24)	4 [3; 5]
26. (a) Management is doing a good job (Unit Management) (^a^ Unit = UCSP/USF), *n* = 635	3.60 (1.22)	4 [3; 5]
26. (b) Management is doing a good job (Hospital Administration). (^a^ hospital = ACeS), *n* = 566	3.03 (1.28)	3 [2; 4]
27. (a) Problem personnel are dealt with constructively by our Administration (Unit Management) (^a^ Unit = UCSP/USF), *n* = 614	3.18 (1.31)	3 [2; 4]
27. (b) Problem personnel are dealt with constructively by our Administration (Hospital Administration). (^a^ hospital = ACeS), *n* = 546	2.79 (1.25)	3 [2; 4]
28. (a) I get adequate, timely info about events that might affect my work, from (Unit Management) (^a^ Unit = UCSP/USF), *n* = 628	3.55 (1.21)	4 [3; 4]
28. (b) I get adequate, timely info about events that might affect my work, from (Hospital Administration). (^a^ hospital = ACeS), *n* = 560	3.19 (1.25)	3 [2; 4]
29. The levels of staffing in this clinical area are sufficient to handle the number of patients, *n* = 638	2.90 (1.48)	3 [2; 4]
***Working conditions* (items 30–32)**		
30. This hospital does a good job of training new personnel. (^a^ hospital = ACeS), *n* = 559	3.52 (1.31)	4 [2; 5]
31. All the necessary information for diagnostic and therapeutic decisions is routinely available to me. *n* = 582	3.41 (1.22)	4 [2; 4]
32. Trainees in my discipline are adequately supervised, *n* = 561	4.04 (1.13)	4 [4; 5]
***Single items* (items 33–36)**		
33. I experience good collaboration with nurses in this clinical area. (^a^ Unit = UCSP/USF), *n* = 624	4.41 (0.95)	5 [4; 5]
34. I experience good collaboration with staff physicians in this clinical area. (^a^ Unit = UCSP/USF), *n* = 619	4.46 (0.85)	5 [4; 5]
35. I experience good collaboration with pharmacists in this clinical area, *n* = 266	3.54 (1.08)	3 [3; 5]
36. Communication breakdowns that lead to delays in delivery of care are common ^b^, *n* = 623	2.93 (1.31)	3 [2; 4]

^a^ References added to the SAQ-Short Form 2006 PT in Google Forms defining equivalence between terms. ^b^ Results obtained after inversion of the item, as described in Section 2.4.

**Table 4 ijerph-19-14344-t004:** Descriptive analysis of the SAQ—Short Form 2006 PT and comparison with demographic and professional characteristics.

Demographic and Professional Characteristics/SAQ-Short Form 2006 PT Domains	Total SAQ-Short Form 2006 PT Score			SAQ-Short Form 2006 PT Scores per Domain																	
				*Team Environment*			*Safety Climate*			*Job Satisfaction*			*Stress Recognition*			*Management Perception*			*Working Conditions*		
	*n*	S	*p*	*n*	S	*p*	*n*	S	*p*	*n*	S	*p*	*n*	S	*p*	*n*	S	*p*	*n*	S	*p*
***Gender***, Med [Q_1_; Q_3_]/M (SD)			**0.019** ^a^			0.119 ^a^			0.182 ^a^			**0.001** ^a^			0.597 ^a^			**0.024** ^c^			0.126 ^a^
*Male*	79	67.4 [54.9; 77.1]		138	79.2 [66.7; 91.7]		135	75 [60.7; 85.7]		133	75 [55; 90]		116	93.8 [75; 100]		106	50.4 (21.8)		115	66.7 [50; 75]	
*Female*	271	72.9 [60.4; 80.6]		470	83.3 [66.7; 95.8]		452	75 [60.7; 89.3]		464	85 [65; 95]		387	87.5 [75; 100]		380	55.9 (22.2)		377	75 [50; 83.3]	
***Age***, r_S_/r_P_	348	−0.014	0.794 ^b^	604	−0.041	0.315 ^b^	583	−0.029	0.487 ^b^	593	0.023	0.578 ^b^	500	−0.166	**<0.001** ^b^	484	0.013	0.775 ^d^	490	−0.003	0.942 ^b^
***Function***, Med [Q_1_; Q_3_]			0.109 ^e^			**<0.001** ^e^			0.165 ^e^			0.360 ^e^			**<0.001** ^e^			0.559 ^e^			**0.016** ^e^
*Technical Assistant*	38	62.8 [53.1; 76.2]		125	79.2 [58.3; 91.7]		120	73.2 [60.7; 85.7]		123	80 [60; 95]		79	75 [37.5; 87.5]		79	61.4 [38.6; 75]		66	58.3 [47.9; 75]	
*Nurse*	128	71.9 [59; 79.9]		198	77.1 [62.5; 91.7]		189	75 [57.1; 85.7]		189	80 [60; 95]		172	87.5 [75; 100]		166	56.8 [38.1; 72.7]		170	66.7 [50; 83.3]	
*Doctor in pre-career training*	49	75.7 [66.7; 79.9]		67	87.5 [75; 91.7]		66	78.6 [64.3; 86.6]		67	90 [75; 95]		65	93.8 [81.3; 100]		57	59.1 [40.9; 70.5]		64	75 [58.3; 89.6]	
*Physicians*	135	70.1 [59.0; 80.6]		218	87.5 [70.8; 95.]		212	75 [57.1; 89.3]		218	80 [60; 95]		187	100 [81.3; 100]		184	52.3 [38.6; 67.6]		192	75 [50; 83.3]	
***Length of service at the*** ***current workplace***, r_S_	350	0.041	0.450 ^b^	608	−0.027	0.500 ^b^	587	−0.020	0.624 ^b^	597	−0.001	0.971 ^b^	503	−0.095	**0.032** ^b^	486	−0.017	0.707 ^b^	492	0.059	0.191 ^b^
***Type of primary care unit***, Med [Q_1_; Q_3_] /M (SD)			**<0.001** ^e^			**<0.001** ^e^			**<0.001** ^e^			<0.001 ^e^			0.601 ^e^			**0.038** ^f^			**<0.001** ^e^
*UCSP*	58	60.4 [47.7; 72.2]		118	66.7 [50; 83.3]		111	60.7 [46.4; 78.6]		114	72.5 [55; 95]		95	93.8 [75; 100]		80	48.9 (21.3)		84	58.3 [41.7; 75]	
*USF-A*	82	73.6 [60.9; 80.6]		141	87.5 [70.8; 95.8]		135	78.6 [64.3; 89.3]		138	80 [60; 95]		116	87.5 [75; 100]		117	56.4 (21.9)		115	75 [50; 83.3]	
*USF-B*	210	72.9 [62.5; 81.3]		349	83.3 [70.8; 91.7]		341	75 [64.3; 89.3]		345	85 [70; 95]		292	93.8 [75; 100]		289	55.6 (22.4)		293	75 [50; 91.7]	

a: Mann–Whitney test; b: linear correlation test for Spearman coefficient; c: *t*-test for two independent samples; d: linear correlation test for Pearson coefficient; e: Kruskal–Wallis test; f: one-way ANOVA test; rS: Spearman coefficient. rP: Pearson coefficient. S: summary measures.

## Data Availability

Data may be shared by request to the authors.

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
