# Peer review of "Safety Climate Evaluation in Primary Health Care: A Cross-Sectional Study"

_ijerph, 2022, doi:10.3390/ijerph192114344_

Round 1
Reviewer 1 Report
General comment:
- I think the study addresses a very important issue and I appreciate that it can help to bring more attention to occupational safety.
- Personally, I would prefer a different data analysis (see below), but at least I cannot find any serious errors in the statistical approaches used here. This is (unfortunately) not always the case with papers I read as a reviewer.
- If you address my points below, I think your work would improve significantly, and I believe they can all be fixed.
Important Comments:
- Section 3.1 introduces the characteristics of the study participants. We also learn that only data from 9.1% of the subjects invited to participate in the study could be used in the analysis. Given that 77% of these subjects are female, it is reasonable to assume that the participant sample is not representative of the overall sample. It would be helpful to know how subjects who chose to participate differ from subjects who did not. Probably not all demographic characteristics of the subjects who did not participate are obtainable. But I am sure that it is possible to obtain many such data from the institutions. Analysis of these data should lead to many interesting findings. For example, is the overall finding that safety is perceived to be rather low driven by the fact that only those who have a problem with safety in their workplace participated in the study? Is this related to the fact that women generally perceive situations as less safe and are more risk averse than men?
- This comment is along similar lines: can you compare your sample with samples from other studies? Perhaps it is typical that, for example, the vast majority of participants in SAQ questionnaires are female? Perhaps it is also normal that so few participants take this survey? If your sample is not representative of the overall sample, but is very similar to other samples that have participated in such studies, you could indeed compare your results to other studies. However, if this is not the case, your results could be driven solely by your sample selection. This, however, would be a major problem for your entire paper.
- I find the presentation of the results in section 3.1 very confusing and unclear. Would it be possible to summarize the results? Maybe not only binary tests should be done. What about regressions? For example, one could code the level of occupation (nurse was probably the lowest and physician the highest?) and add that as an explanatory variable. Then you should see if, for example, the higher the position, the less people feel safe (or something like that). But now I don't know who are the people who are least satisfied. Is it the women? Is it the elderly? Is it the nurses? And the same question applies to each domain....
- That's also part of the comments above: I found in line 518 the sentence "The sample obtained was representative of the population studied." Is that correct? I found no evidence of this in section 3.1.
- Section 4.1. also presents too much detail about individual findings and individual items. You should give the reader something that is easy to digest. Tell the reader what the overall findings are. What are the lessons that can be learned? Summarize your findings and fit them into an overall picture.
- In general, I would suggest putting all the analysis of each item in the appendix (at most) and really focus more on the main domains. Otherwise it gets very confusing and the reader gets lost in all the details. You show that the Cronbach alphas within each domain are high enough. So there is absolutely no need to go into detail about each item. Minor
Tell the reader what the overall findings are. What are the lessons that can be learned? Summarize your findings and fit them into an overall picture.
- In general, I would suggest putting all the analysis of each point in the appendix at most and really focus more on the main areas. Otherwise it gets very confusing and the reader gets lost in all the details. They show that the Cronbach alphas within each domain are high enough. So there is absolutely no need to go into detail about each item.
Minor comments:
- In line 99 and in line 100 you say twice that inexperienced workers were excluded.
- In line 190, it should be "They didn't meet" and not "didn't met".
- In lines 216-222, why is the font in bold smaller than the usual font and partial?
Author Response
Dear Professor,
We would like to thank you for the review of the article and suggestions made, so important for us authors. We´ve read your comments and wrote a small text in reply. We please ask for a new review. We are truly thankfull for the contributions made.
Important Comments:
Comment 1
Section 3.1 introduces the characteristics of the study participants. We also learn that only data from 9.1% of the subjects invited to participate in the study could be used in the analysis. Given that 77% of these subjects are female, it is reasonable to assume that the participant sample is not representative of the overall sample. It would be helpful to know how subjects who chose to participate differ from subjects who did not. Probably not all demographic characteristics of the subjects who did not participate are obtainable. But I am sure that it is possible to obtain many such data from the institutions. Analysis of these data should lead to many interesting findings. For example, is the overall finding that safety is perceived to be rather low driven by the fact that only those who have a problem with safety in their workplace participated in the study? Is this related to the fact that women generally perceive situations as less safe and are more risk averse than men?
In Table 1 and in Appendix B is presented the descriptive analysis of the sample and the population, respectively, where can be seen that the sample is representative of the population studied. The author´s tried to find data related to the gender of the professionais but it wasn´t avaluable for the period in study 1.2022. In Portugal female gender professionals clearly outnumber male gender professionals. For instance, since 2010, female doctors have outnumbered male doctors and that difference has been increasing year after year (https://www.pordata.pt/db/portugal/ambiente+de+consulta/tabela). This is also seen in medical schools.
A study implemented in Slovenia - Klemenc-Ketis et al. BMC Health Services Research (2018) 18:767 https://doi.org/10.1186/s12913-018-3594-8 - also registered a women participation of 88.3%.
A study implemented in Jordania -Khamaiseh A; Al-Twalbeh D; Al-Ajlouni K. Patient safety culture in Jordanian primary health-care centres as perceived by nurses: a cross-sectional study. East Mediterr Health J. 2020;26(10):1242–1250. https://doi.org/10.26719/emhj.20.044- also found a women participation of 85.7%.
Comment 2
This comment is along similar lines: can you compare your sample with samples from other studies? Perhaps it is typical that, for example, the vast majority of participants in SAQ questionnaires are female? Perhaps it is also normal that so few participants take this survey? If your sample is not representative of the overall sample, but is very similar to other samples that have participated in such studies, you could indeed compare your results to other studies. However, if this is not the case, your results could be driven solely by your sample selection. This, however, would be a major problem for your entire paper.
We´ve read and discuss different papers but comparison is very limited because there are major differences in the population studied (most studies were implemented in hospitals) or in the SAQ questionnaires used (for instance combining items in different dimensions). (Please consider 4.2. Comparison with Prior Work).
Comment 3
I find the presentation of the results in section 3.1 very confusing and unclear. Would it be possible to summarize the results? Maybe not only binary tests should be done. What about regressions? For example, one could code the level of occupation (nurse was probably the lowest and physician the highest?) and add that as an explanatory variable. Then you should see if, for example, the higher the position, the less people feel safe (or something like that). But now I don't know who are the people who are least satisfied. Is it the women? Is it the elderly? Is it the nurses? And the same question applies to each domain....
In order to optimize the description of the results presented in 3.1. Descriptive analysis of the sample these should be inserted in a table? This was the first study to implement SAQ in PHC in Portugal and there are many interesting data still to explore. We felt like this study could open new research paths for other studies.
Comment 4
That's also part of the comments above: I found in line 518 the sentence "The sample obtained was representative of the population studied." Is that correct? I found no evidence of this in section 3.1.
In Table 1 and in Appendix B is presented the descriptive analysis of the sample and the population, respectively, in order to justify the sentence.
Comment 5
Section 4.1. also presents too much detail about individual findings and individual items. You should give the reader something that is easy to digest. Tell the reader what the overall findings are. What are the lessons that can be learned? Summarize your findings and fit them into an overall picture. (This section was improved)
Comment 6
In general, I would suggest putting all the analysis of each item in the appendix (at most) and really focus more on the main domains. Otherwise it gets very confusing and the reader gets lost in all the details. You show that the Cronbach alphas within each domain are high enough. So there is absolutely no need to go into detail about each item. Minor
Comment 7
Tell the reader what the overall findings are. What are the lessons that can be learned? Summarize your findings and fit them into an overall picture. (This section was improved)
Minor comments
- In line 99 and in line 100 you say twice that inexperienced workers were excluded.
(Done)
- In line 190, it should be "They didn't meet" and not "didn't met".
(Done)
- In lines 216-222, why is the font in bold smaller than the usual font and partial?
(Done)

Reviewer 2 Report
Congratulations on the article. The topic is very interesting and the results are elaborate.
I suggest improving the introduction and especially the discussion. The discussion must be completely reformulated.
Comments:
Abstract: define PHC the first time it appears(page 1, line 20).
Introduction
Page 1, line 35 a 40. The bibliographic citations are missing.
I suggest expanding the introduction bibliographically, for example detailing security incidents at PHC and the results of these studies in other PHC contexts. Only 4 quotes from a widely studied topic appear. Also, two of them (1,3) are references of the questionnaires used.
Page 2, line 55-62. This SAQ instrument had never been used at PHC? If it has been used, the results could be detailed (critical incidents detected etc..).
Do not separate into two paragraphs, the first serves as an introduction to the evaluation.
Proposal:
“Assessing the safety climate through questionnaires is a practical tool that allows organizations to act proactively and plan improvement strategies, evaluating the impact and effectiveness of implemented actions [1]. Sexton et al.'s Safety Attitudes Questionnaire (SAQ) is one of the most widely used tools, has been adapted for use in intensive care units, operating rooms, general inpatientand ambulatory clinics [1, 3] (Appendix A). It has been extensively tested, is considered psychometrically valid, and allows the assessment of health professionals' perceptions and attitudes related to safety [1]. The original version was developed in 2006 by BryanSexton, Eric Thomas and Bob Helmreich (University of Texas) based on the Intensive Care Unit Management Attitudes Questionnaire, which in turn originated from the Flight Man-61 agement Attitudes Questionnaire, widely used in commercial aviation [3].
Also, the authors could consider detailing the instrument in the Methodology section, not in the introduction. It is the first time that I see an appendix of an instrument in an introduction.
Page 2, line 76-79. The objectives should be unified in a single paragraph (main objective and secondary objective?) and they should not have bibliographic quotes.
Materials and Methods
Page 2, line 83. Study design. Delete information about data collection. Move it to the Data Collection section.
Review the wording. Methodology is always written in past tense.
Page 2-3, line 96-101. Review the inclusion and exclusion criteria of the participants. I think it is a wording problem as they seem to be the same. It is not understood. “(as defined [3])” (page 3. Line 101).
Page 3, line 104. Be careful, the deadline dates for the questionnaire appear to be different; August 9 or 5?
Page 3, line 102 I suggest 2.3 Variables and instruments. Delete all data collection information (section 2.5).
Briefly explain the concept of “snowball approach” (page 3, line 105), I understand it is a non-probabilistic sampling. The concept “snowball approach” is more typical of qualitative studies.
Page 4, line 154. Cronbach’s alpha put two decimals 0.70.
Page 4. line 142-143 . It doesn´t matter if I go from Google Form to Excell and then to SPSS, it matters which instrument was used to analyze the data (SPSS). I would give this information at the end of this section and in summary form, along with statistical significance.
Page 4, line 165-166. It does not make much sense to explain here how the graph in figure 1 was constructed.
Page 4, section Data Collection. Only talk about data collection. The variables must appear in section 2.3. Do not repeat the objective.
I would eliminate section 2.7 (Page 4, line 182) of the Methodology. Actually, I would include this idea in the introduction.
Results
Beware, the confusion between the inclusion and exclusion criteria reappears. It is normal that no one was excluded as they did not actually meet the inclusion criteria (Page4, line 193). Review wording.
Page 5, section 3.2. I would present th consistency data first (Cronbach´s alpha , α) and the general data(page 10, line 249-264, Table 3), and then the data in table 2 and figure 1. These data are more specific.
Page 5, line 207-222. I would simplify the wording since the data already appears in table 2.
Presenting so any results makes monitoring difficult. Do you consider the information in figure 2 essential (page 13, line 337-364)? By not describing the contexts separately, it seems to provide very local information.…
Discussion
In the discussion you should not cite the tables or figures, only the findings, and discuss them using the literature. You re-describe the results, but do not discuss them, it is a summary.
You must investigate the reasons for your results from the literature or from the explanations you can provide. Nor should they include all the statistical data found.
This section should be reformulated. In addition, the instrument has been widely used (as they have been named in the introduction), so you should be able to find many points of discussion with other research from other countries or contexts.
This is actually the section you must improve for your work to be publishable.
The bibliography is very limited.
Author Response
Dear Professor,
We would like to thank you for the review of the article and suggestions made, so important for us authors. We´ve read your comments and wrote a small text in reply. We please ask for a new review. We are truly thankfull for the contributions made.
Comment 1
I suggest improving the introduction and especially the discussion. The discussion must be completely reformulated.
Comment 2
Abstract: define PHC the first time it appears(page 1, line 20). (Done)
Comment 3
Introduction
- Page 1, line 35 a 40. The bibliographic citations are missing. (Done)
- I suggest expanding the introduction bibliographically, for example detailing security incidents at PHC and the results of these studies in other PHC contexts. Only 4 quotes from a widely studied topic appear. Also, two of them (1,3) are references of the questionnaires used.(Done)
- Page 2, line 55-62. This SAQ instrument had never been used at PHC? If it has been used, the results could be detailed (critical incidents detected etc..). (This item was improved in discussion)
- Do not separate into two paragraphs, the first serves as an introduction to the evaluation. Proposal: “Assessing the safety climate through questionnaires is a practical tool that allows organizations to act proactively and plan improvement strategies, evaluating the impact and effectiveness of implemented actions [1]. Sexton et al.'s Safety Attitudes Questionnaire (SAQ) is one of the most widely used tools, has been adapted for use in intensive care units, operating rooms, general inpatientand ambulatory clinics [1, 3] (Appendix A). It has been extensively tested, is considered psychometrically valid, and allows the assessment of health professionals' perceptions and attitudes related to safety [1]. The original version was developed in 2006 by BryanSexton, Eric Thomas and Bob Helmreich (University of Texas) based on the Intensive Care Unit Management Attitudes Questionnaire, which in turn originated from the Flight Man-61 agement Attitudes Questionnaire, widely used in commercial aviation [3]. (Done)
- Also, the authors could consider detailing the instrument in the Methodology section, not in the introduction. It is the first time that I see an appendix of an instrument in an introduction. (Done)
- Page 2, line 76-79. The objectives should be unified in a single paragraph (main objective and secondary objective?) and they should not have bibliographic quotes. (Done)
Comment 4
Materials and Methods
- Page 2, line 83. Study design. Delete information about data collection. Move it to the Data Collection section. (Done)
- Review the wording. Methodology is always written in past tense. (Done)
- Page 2-3, line 96-101. Review the inclusion and exclusion criteria of the participants. I think it is a wording problem as they seem to be the same. It is not understood. “(as defined [3])” (page 3. Line 101). (Done)
- Page 3, line 104. Be careful, the deadline dates for the questionnaire appear to be different; August 9 or 5? (Done)
- Page 3, line 102 I suggest 2.3 Variables and instruments. Delete all data collection information (section 2.5). (Done)
- Briefly explain the concept of “snowball approach” (page 3, line 105), I understand it is a non-probabilistic sampling. The concept “snowball approach” is more typical of qualitative studies. (Done)
- Page 4, line 154. Cronbach’s alpha put two decimals 0.70. (Done)
- Page 4. line 142-143 . It doesn´t matter if I go from Google Form to Excell and then to SPSS, it matters which instrument was used to analyze the data (SPSS). I would give this information at the end of this section and in summary form, along with statistical significance. (Done)
- Page 4, line 165-166. It does not make much sense to explain here how the graph in figure 1 was constructed. (Done)
- Page 4, section Data Collection. Only talk about data collection. The variables must appear in section 2.3. Do not repeat the objective. (Done)
- I would eliminate section 2.7 (Page 4, line 182) of the Methodology. Actually, I would include this idea in the introduction. (Done)
Comment 5
Results
- Beware, the confusion between the inclusion and exclusion criteria reappears. It is normal that no one was excluded as they did not actually meet the inclusion criteria (Page4, line 193). Review wording. (Done)
- Page 5, section 3.2. I would present th consistency data first (Cronbach´s alpha , α) and the general data(page 10, line 249-264, Table 3), and then the data in table 2 and figure 1. These data are more specific. (Done)
- Page 5, line 207-222. I would simplify the wording since the data already appears in table 2. (Done)
- Presenting so any results makes monitoring difficult. Do you consider the information in figure 2 essential (page 13, line 337-364)? By not describing the contexts separately, it seems to provide very local information.…
Safety Climate varies according to specific contexts and it is possible to see that analysing figure 2. We believe this figure raises new questions and possible lines of investigation. It carries an intrinsic value and appeals for the replication of the study in other places/countries. We would like so much to keep it if possible.
Comment 6
Discussion
- In the discussion you should not cite the tables or figures, only the findings, and discuss them using the literature. You re-describe the results, but do not discuss them, it is a summary. (Done)
- You must investigate the reasons for your results from the literature or from the explanations you can provide. Nor should they include all the statistical data found. (This section was improved)
- This section should be reformulated. In addition, the instrument has been widely used (as they have been named in the introduction), so you should be able to find many points of discussion with other research from other countries or contexts. (This section was improved)
This is actually the section you must improve for your work to be publishable. (This section was improved)
- The bibliographyis very limited. (Done)
